# DreamerPro: Reconstruction-Free Model-Based Reinforcement Learning with Prototypical Representations

## Abstract

Top-performing Model-Based Reinforcement Learning (MBRL) agents, such as Dreamer, learn the world model by reconstructing the image observations. Hence, they often fail to discard task-irrelevant details and struggle to handle visual distractions. To address this issue, previous work has proposed to contrastively learn the world model, but the performance tends to be inferior in the absence of distractions. In this paper, we seek to enhance robustness to distractions for MBRL agents. Specifically, we consider incorporating prototypical representations, which have yielded more accurate and robust results than contrastive approaches in computer vision. However, it remains elusive how prototypical representations can benefit temporal dynamics learning in MBRL, since they treat each image independently without capturing temporal structures. To this end, we propose to learn the prototypes from the recurrent states of the world model, thereby distilling temporal structures from past observations and actions into the prototypes. The resulting model, DreamerPro, successfully combines Dreamer with prototypes, making large performance gains on the DeepMind Control suite when there are complex background distractions, while maintaining similar performance as Dreamer in the standard setting.

## 1 Introduction

Model-Based Reinforcement Learning (MBRL, Sutton & Barto, 2018; Sutton, 1991) provides a solution to many problems in contemporary reinforcement learning. It improves sample efficiency by training a policy through simulations of a learned world model. Learning a world model also provides a way to efficiently represent experience data as general knowledge simulatable and reusable in arbitrary downstream tasks. In addition, it allows accurate and safe decisions via planning.

Among recent advances in image-based MBRL, Dreamer is particularly notable as the first MBRL model outperforming popular model-free RL algorithms with better sample efficiency in both continuous control (Hafner et al., 2020) and discrete control (Hafner et al., 2021). Unlike some previous model-based RL methods (Kaiser et al., 2019), it learns a world model that can be rolled out in a compact latent representation space instead of the high-dimensional observation space. Also, policy learning can be done efficiently via backpropagation through the differentiable dynamics model.

In image-based RL, the key problem is to learn low-dimensional state representation and, in the model-based case, also its forward model. Although we can learn such representation directly by maximizing the rewards (Schrittwieser et al., 2020), it is usually very slow to do this due to the reward sparsity. Instead, it is more practical to introduce auxiliary tasks providing richer learning signal to facilitate representation learning without reward (or with sparse reward) (Sutton et al., 2011; Jaderberg et al., 2016). Dreamer achieves this by learning the representation and the dynamics model in a way to reduce the reconstruction error of the observed sequences. However, reconstruction-based representation learning has limitations. First, it is computationally expensive to reconstruct the high-dimensional inputs, especially in models like Dreamer that needs to reconstruct long-range videos. Second, it wastes the representation capacity to learn even the visual signals that are irrelevant to the task or unpredictable such as noisy background (Burda et al., 2018). Thus, in MBRL it is of particular interest to realize a version of Dreamer without reconstruction.

Recently, there have been remarkable advances in reconstruction-free representation learning in reinforcement learning (Laskin et al., 2020a;b; Yarats et al., 2021c). The currently dominant approach is via contrastive learning. This approach requires pair-wise comparisons to push apart different instances while pulling close an instance and its augmentation. Therefore, this method usually requires a large batch size (so computationally expensive) to perform accurately and robustly. An alternative is the clustering-based or prototype-based approach (Caron et al., 2020). By learning a set of clusters represented by prototypes, it replaces the instance-wise comparison by a comparison to the clusters and thereby avoids the problems of contrastive learning. This approach is shown to perform more accurately and robustly in many applications (Caron et al., 2020; 2021; Yarats et al., 2021b) than the contrastive method while also alleviating the need for maintaining a large batch size. The prototype structure can also be used to implement an exploration method (Yarats et al., 2021b).

However, for reconstruction-free MBRL only the contrastive approach like Temporal Predictive Coding (TPC, Nguyen et al., 2021) has been proposed so far. While TPC consistently outperforms DREAMER in the noisy background settings, for standard DeepMind Control suite (Tassa et al., 2018) it showed quite inconsistent results by performing severely worse than DREAMER on some tasks. Therefore, we hypothesize that this inconsistent behavior may be fixed if the robustness and accuracy of the prototypical representations can be realized in MBRL and further improved with the support of temporal information.

In this paper, we propose a reconstruction-free MBRL agent, called DREAMERPRO, by combining the prototypical representation learning with temporal dynamics learning. Similar to SwAV (Caron et al., 2020), by encouraging uniform cluster assignment across the batch, we implicitly pull apart the embeddings of different observations. Additionally, we let the temporal latent state to 'reconstruct' the cluster assignment of the observation, thereby relieving the world model from modeling low-level details. We evaluate our model on the standard setting of DeepMind Control suite, and also on a natural background setting, where the background is replaced by natural videos irrelevant to the task. The results show that the proposed model consistently outperforms previous methods.

The contributions of the paper are (1) the first reconstruction-free MBRL agent based on the prototypical representation and its temporal dynamics and (2) the demonstration of the consistently improved accuracy and robustness of the proposed model in comparison to a contrastive reconstruction-free MBRL agent and Dreamer for both standard and natural background DMC tasks.

## 2 PRELIMINARIES

In this section, we briefly introduce the world model and learning algorithms used in DREAMERV2 (Hafner et al., 2021) which our model builds upon. To indicate the general DREAMER framework (Hafner et al., 2020; 2021), we omit its version number in the rest of the paper.

### 2.1 RECONSTRUCTION-BASED WORLD MODEL LEARNING

DREAMER learns a recurrent state-space model (RSSM, Hafner et al., 2019) to predict forward dynamics and rewards in partially observable environments. At each time step $t$, the agent receives an image observation $o_t$ and a scalar reward $r_t$ (obtained by previous actions $a_{<t}$). The agent then chooses an action $a_t$ based on its policy. The RSSM models the observations, rewards, and transitions through a probabilistic generative process:

$$p(o_{1:T}, r_{1:T} \mid a_{1:T}) = \int \prod_{t=1}^{T} p(o_t \mid s_{\leq t}, a_{<t}) \, p(r_t \mid s_{\leq t}, a_{<t}) \, p(s_t \mid s_{<t}, a_{<t}) \, \mathrm{d}s_{1:T} \quad (1)$$

$$= \int \prod_{t=1}^{T} p(o_t \mid h_t, s_t) \, p(r_t \mid h_t, s_t) \, p(s_t \mid h_t) \, \mathrm{d}s_{1:T} \,, \quad (2)$$

where the latent variables $s_{1:T}$ are the agent states, and $h_t = \mathrm{GRU}(h_{t-1}, s_{t-1}, a_{t-1})$ is a deterministic encoding of $s_{<t}$ and $a_{<t}$. To infer the agent states from past observations and actions, a variational encoder is introduced:

$$q(s_{1:T} \mid o_{1:T}, a_{1:T}) = \prod_{t=1}^{T} q(s_t \mid s_{<t}, a_{<t}, o_t) = \prod_{t=1}^{T} q(s_t \mid h_t, o_t) \,. \quad (3)$$

The training objective is to maximize the evidence lower bound (ELBO):

$$\mathcal{J}_{\text{DREAMER}} = \sum_{t=1}^{T} \mathbb{E}_q [\underbrace{\log p(o_t \mid h_t, s_t)}_{\mathcal{J}_{\text{O}}^t} + \underbrace{\log p(r_t \mid h_t, s_t)}_{\mathcal{J}_{\text{R}}^t} - \underbrace{D_{\text{KL}}(q(s_t \mid h_t, o_t) \parallel p(s_t \mid h_t))}_{\mathcal{J}_{\text{KL}}^t}] . \quad (4)$$

### 2.2 POLICY LEARNING BY LATENT IMAGINATION

DREAMER interleaves policy learning with world model learning. During policy learning, the world model is fixed, and an actor and a critic are trained cooperatively from the latent trajectories imagined by the world model. Specifically, the imagination starts at each non-terminal state $\hat{z}_t = [h_t, s_t]$ encountered during world model learning. Then, at each imagination step $t' \geq t$, an action is sampled from the actor's stochastic policy: $\hat{a}_{t'} \sim \pi(\hat{a}_{t'} \mid \hat{z}_{t'})$. The corresponding reward $\hat{r}_{t'+1}$ and next state $\hat{z}_{t'+1}$ are predicted by the learned world model. Given the imagined trajectories, the actor improves its policy by maximizing the $\lambda$-return (Sutton & Barto, 2018; Schulman et al., 2018) plus an entropy regularizer that encourages exploration, while the critic is trained to approximate the $\lambda$-return through a squared loss.

## 3 DREAMERPRO

To compute the DREAMER training objective, more specifically $\mathcal{J}_{\text{O}}^t$ in Equation 4, a decoder is required to reconstruct the image observation $o_t$ from the state $z_t = [h_t, s_t]$. Because this reconstruction loss operates in pixel space where all pixels are weighted equally, DREAMER tends to allocate most of its capacity to modeling complex visual patterns that cover a large pixel area (e.g., backgrounds). This leads to poor task performance when those visual patterns are task irrelevant, as shown in previous work (Nguyen et al., 2021).

Fortunately, during policy learning, what we need is accurate reward and next state prediction, which are respectively encouraged by $\mathcal{J}_{\text{R}}^t$ and $\mathcal{J}_{\text{KL}}^t$. In other words, the decoder is not required for policy learning. The main purpose of having the decoder and the associated loss $\mathcal{J}_{\text{O}}^t$, as shown in DREAMER, is to learn meaningful representations that cannot be obtained by $\mathcal{J}_{\text{R}}^t$ and $\mathcal{J}_{\text{KL}}^t$ alone.

The above observations motivate us to improve robustness to visual distractions by replacing the reconstruction-based representation learning in DREAMER with reconstruction-free methods. For this, we take inspiration from recent developments in self-supervised image representation learning, which can be divided into contrastive (van den Oord et al., 2019; Chen et al., 2020; He et al., 2020) and non-contrastive (Grill et al., 2020; Caron et al., 2020) methods. We prefer non-contrastive methods as they can be applied to small batch sizes. This can speed up both world model learning and policy learning (in wall clock time). Therefore, we propose to combine DREAMER with the prototypical representations used in SWAV (Caron et al., 2020), a top-performing non-contrastive representation learning method. We name the resulting model DREAMERPRO, and provide the model description in the following.

DREAMERPRO uses the same policy learning algorithm as DREAMER, but learns the world model without reconstructing the observations. This is achieved by clustering the observation into a set of $K$ trainable prototypes $\{c_1, \ldots, c_K\}$, and then predicting the cluster assignment from the state as well as an augmented view of the observation. See Figure 1 for an illustration.

Concretely, given a sequence of observations $o_{1:T}$ sampled from the replay buffer, we obtain two augmented views $o_{1:T}^{(1)}, o_{1:T}^{(2)}$ by applying random shifts (Laskin et al., 2020b; Yarats et al., 2021c) with bilinear interpolation (Yarats et al., 2021a). We ensure that the augmentation is consistent across time steps. Each view $i \in \{1, 2\}$ is fed to the RSSM to obtain the states $z_{1:T}^{(i)}$. To predict the cluster assignment from $z_t^{(i)}$, we first apply a linear projection followed by $\ell_2$-normalization to obtain a vector $x_t^{(i)}$ of the same dimension as the prototypes, and then take a softmax over the dot products of $x_t^{(i)}$ and all the prototypes:

$$(u_{t,1}^{(i)}, \ldots, u_{t,K}^{(i)}) = \text{softmax}\left(\frac{x_t^{(i)} \cdot c_1}{\tau}, \ldots, \frac{x_t^{(i)} \cdot c_K}{\tau}\right) . \quad (5)$$

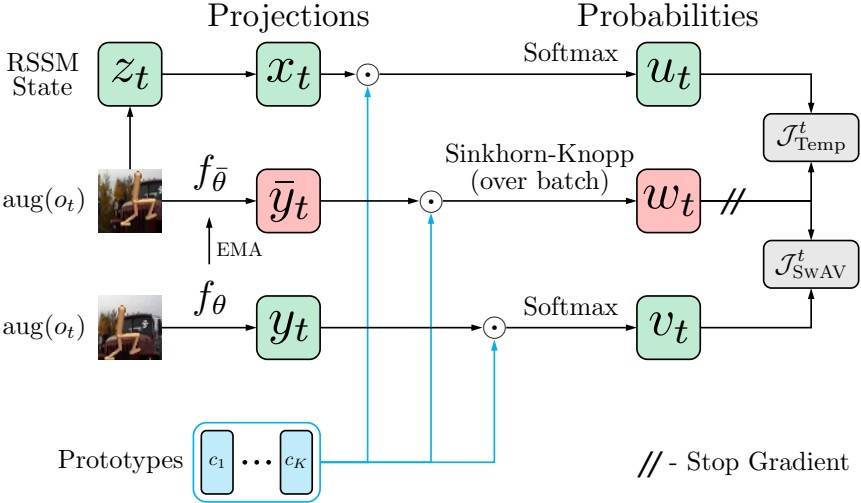

Figure 1: DREAMERPRO learns the world model through online clustering, eliminating the need for reconstruction. At each time step $t$, it first compares the observation to a set of trainable prototypes $\{c_1, \ldots, c_K\}$ to obtain the target cluster assignment $w_t$. Then, it predicts this target from both the world model state $z_t$ and another augmented view of the observation (each aug($o_t$) denotes an independent application of data augmentation). The predictions are improved by optimizing the two objective terms, $\mathcal{J}^t_{\text{Temp}}$ and $\mathcal{J}^t_{\text{SwAV}}$, respectively, where the first term crucially distills temporal structures from $z_t$ into the prototypes.

Here, $u^{(i)}_{t,k}$ is the predicted probability that state $z^{(i)}_t$ maps to cluster $k$, $\tau$ is a temperature parameter, and the prototypes $\{c_1, \ldots, c_K\}$ are also $\ell_2$-normalized.

Analogously, to predict the cluster assignment from an augmented observation $o^{(i)}_t$, we feed it to a convolutional encoder (shared with the RSSM), apply a linear projection followed by $\ell_2$-normalization, and obtain a vector $y^{(i)}_t$. We summarize this process as: $y^{(i)}_t = f_\theta(o^{(i)}_t)$, where $\theta$ collectively denotes the parameters of the convolutional encoder and the linear projection layer. The prediction probabilities are again given by a softmax:

$$(v^{(i)}_{t,1}, \ldots, v^{(i)}_{t,K}) = \text{softmax}\left( \frac{y^{(i)}_t \cdot c_1}{\tau}, \ldots, \frac{y^{(i)}_t \cdot c_K}{\tau} \right), \qquad (6)$$

where $v^{(i)}_{t,k}$ is the predicted probability that observation $o^{(i)}_t$ maps to cluster $k$.

To obtain the targets for the above two predictions (i.e., Equations 5 and 6), we apply the Sinkhorn-Knopp algorithm (Cuturi, 2013) to the cluster assignment scores computed from the output of a momentum encoder $f_{\bar{\theta}}$ (He et al., 2020; Grill et al., 2020; Caron et al., 2021), whose parameters $\bar{\theta}$ are updated using the exponential moving average of $\theta$: $\bar{\theta} \leftarrow (1 - \eta)\bar{\theta} + \eta\theta$. For each observation $o^{(i)}_t$, the scores are given by the dot products $(\bar{y}^{(i)}_t \cdot c_1, \ldots, \bar{y}^{(i)}_t \cdot c_K)$, where $\bar{y}^{(i)}_t = f_{\bar{\theta}}(o^{(i)}_t)$ is the momentum encoder output. The Sinkhorn-Knopp algorithm is applied to the two augmented batches $\{o^{(1)}_{1:T}\}, \{o^{(2)}_{1:T}\}$ separately to encourage uniform cluster assignment within each augmented batch and avoid trivial solutions. We specifically choose the number of prototypes $K = B \times T$, where $B$ is the batch size, so that the observation embeddings are implicitly pushed apart from each other. The outcome of the Sinkhorn-Knopp algorithm is a set of cluster assignment targets $(w^{(i)}_{t,1}, \ldots, w^{(i)}_{t,K})$ for each observation $o^{(i)}_t$.

Now that we have the cluster assignment predictions and targets, the representation learning objective is simply to maximize the prediction accuracies:

$$\mathcal{J}_{\text{SwAV}}^t = \frac{1}{2} \sum_{k=1}^{K} \left( w_{t,k}^{(1)} \log v_{t,k}^{(2)} + w_{t,k}^{(2)} \log v_{t,k}^{(1)} \right) \ , \tag{7}$$

$$\mathcal{J}_{\text{Temp}}^t = \frac{1}{2} \sum_{k=1}^{K} \left( w_{t,k}^{(1)} \log u_{t,k}^{(1)} + w_{t,k}^{(2)} \log u_{t,k}^{(2)} \right) \ . \tag{8}$$

Here, $\mathcal{J}_{\text{SwAV}}^t$ improves prediction from an augmented view. This is the same loss as used in SwAV (Caron et al., 2020), and is shown to induce useful features for static images. However, it ignores the temporal structure which is crucial in reinforcement learning. Hence, we add a second term, $\mathcal{J}_{\text{Temp}}^t$, that improves prediction from the state of the same view. This has the effect of making the prototypes close to the states that summarize the past observations and actions, thereby distilling temporal structure into the prototypes. From another perspective, $\mathcal{J}_{\text{Temp}}^t$ is similar to $\mathcal{J}_{\text{O}}^t$ in the sense that we are now 'reconstructing' the cluster assignment of the observation instead of the observation itself. This frees the world model from modeling complex visual details, allowing more capacity to be devoted to task-relevant features.

The overall world model learning objective for DREAMERPRO can be obtained by replacing $\mathcal{J}_{\text{O}}^t$ in Equation 4 with $\mathcal{J}_{\text{SwAV}}^t + \mathcal{J}_{\text{Temp}}^t$:

$$\mathcal{J}_{\text{DREAMERPRO}} = \sum_{t=1}^{T} \mathbb{E}_q[\mathcal{J}_{\text{SwAV}}^t + \mathcal{J}_{\text{Temp}}^t + \mathcal{J}_{\text{R}}^t - \mathcal{J}_{\text{KL}}^t] \ , \tag{9}$$

where $\mathcal{J}_{\text{R}}^t$ and $\mathcal{J}_{\text{KL}}^t$ are now averaged over the two augmented views.

## 4 EXPERIMENTS

**Environments.** We evaluate our model and the baselines on six image-based continuous control tasks from the DeepMind Control (DMC) suite (Tassa et al., 2018). We choose the set of tasks based on those considered in PLANET (Hafner et al., 2019). Specifically, we replace Cartpole Swingup and Walker Walk with their more challenging counterparts, Cartpole Swingup Sparse and Walker Run, and keep the remaining tasks. In addition to the standard setting, we also consider a natural background setting (Zhang et al., 2021; Nguyen et al., 2021), where the background is replaced by task-irrelevant natural videos randomly sampled from the 'driving car' class in the Kinetics 400 dataset (Kay et al., 2017). Following TPC (Nguyen et al., 2021), we use two separate sets of background videos for training and evaluation. Hence, the natural background setting tests generalization to unseen distractions. We note that the recently released Distracting Control Suite (DCS, Stone et al., 2021) serves a similar purpose. However, the background distractions in DCS seem less challenging, as there are fewer videos and the ground plane is made visible for most tasks. In our preliminary experiments, our model and all the baselines achieved close to zero returns on Cartpole Swingup Sparse in the natural background setting. We therefore switch back to Cartpole Swingup in this setting.

**Baselines.** Our main baselines are DREAMER (Hafner et al., 2021), DREAMING (Okada & Taniguchi, 2021), and TPC (Nguyen et al., 2021), the state-of-the-art for reconstruction-based and reconstruction-free MBRL. In particular, TPC has shown better performance than CVRL (Ma et al., 2020), DBC (Zhang et al., 2021), and CURL (Laskin et al., 2020a) on the same datasets. The recently proposed PSE (Agarwal et al., 2021) has demonstrated impressive results on DCS. However, it is only shown to work in the model-free setting and requires a pretrained policy, while our model learns both the world model and the policy from scratch.

**Implementation details.** We implement our model and DREAMING based on a newer version of DREAMER[1], while the official implementation of TPC[2] is based on an older version. For fair

---

[1] https://github.com/danijar/dreamerv2/tree/e783832f01b2c845c195587158c4e129edabaebb

[2] https://github.com/VinAIResearch/TPC-tensorflow

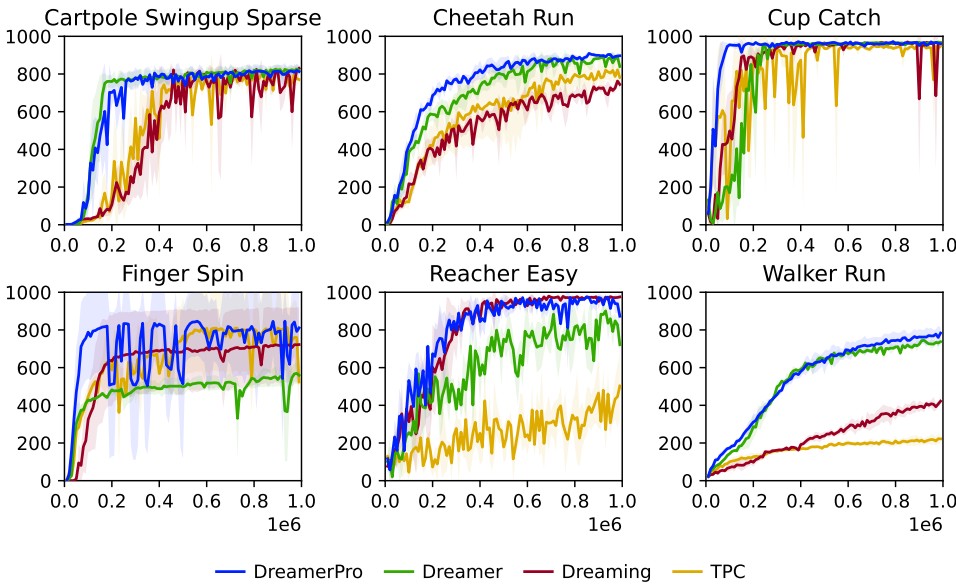

Figure 2: Performance curves in standard DMC. DREAMERPRO is the only model that is comparable or better than DREAMER on all tasks. In particular, DREAMERPRO greatly outperforms DREAMER on Finger Spin and Reacher Easy, and achieves better data efficiency on Cup Catch.

Table 1: Final performance in standard DMC.

| Task | DREAMER | DREAMING | TPC | DREAMERPRO |
|---|---|---|---|---|
| Cartpole Swingup Sparse | $\mathbf{820 \pm 23}$ | $\mathbf{830 \pm 12}$ | $770 \pm 9$ | $\mathbf{813 \pm 32}$ |
| Cheetah Run | $840 \pm 74$ | $745 \pm 18$ | $782 \pm 82$ | $\mathbf{897 \pm 8}$ |
| Cup Catch | $\mathbf{967 \pm 3}$ | $965 \pm 13$ | $948 \pm 7$ | $961 \pm 10$ |
| Finger Spin | $559 \pm 54$ | $722 \pm 197$ | $524 \pm 127$ | $\mathbf{811 \pm 232}$ |
| Reacher Easy | $721 \pm 51$ | $\mathbf{975 \pm 2}$ | $503 \pm 185$ | $873 \pm 127$ |
| Walker Run | $\mathbf{737 \pm 26}$ | $422 \pm 25$ | $222 \pm 29$ | $\mathbf{784 \pm 28}$ |

comparison, we re-implement TPC based on the newer version. We adopt the default values for the DREAMER hyperparameters, except that we use continuous latents and `tanh_normal` as the distribution output by the actor. We find these changes improve DREAMER's performance in the standard DMC, and therefore use these values for all models in both the standard and the natural background setting. Following TPC, we increase the weight of the reward loss $\mathcal{J}_R^t$ to 1000 for all models in the natural background setting to further encourage extraction of task-relevant information. While in the original TPC, this weight is chosen separately for each task from $\{100, 1000\}$, we find the weight of 1000 works consistently better in our re-implementation, which also obtains better results than reported in the original paper. We use the default batch size of 50 for DREAMER, DREAMING, and DREAMERPRO. The batch size for TPC is chosen to be 150, so that it has similar wall clock training time as DREAMERPRO.

**Evaluation protocol.** For each task, we train each model for 1M environment steps (equivalent to 500K actor steps, as the action repeat is set to 2). The evaluation return is computed every 10K steps, and averaged over 10 episodes. In all figures and tables, the mean and standard deviation are computed from 3 independent runs.

## 4.1 PERFORMANCE IN STANDARD DMC

We show the performance curves in Figure 2 and the final performance in Table 1 for the standard setting. DREAMERPRO is the only model that achieves comparable or even better performance than DREAMER on all tasks. Notably, DREAMERPRO outperforms DREAMER by a large margin on

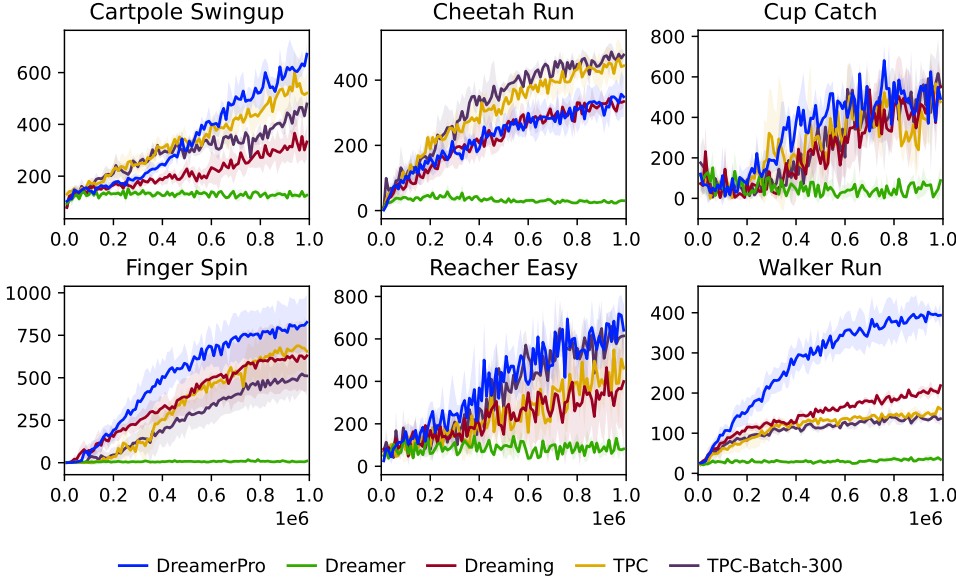

Figure 3: Performance curves in natural background DMC. DREAMERPRO significantly outperforms all baselines on Cartpole Swingup, Finger Spin, and Walker Run, while DREAMER completely fails on all tasks.

Table 2: Final performance in natural background DMC.

| Task | DREAMER | DREAMING | TPC | TPC-Batch-300 | DREAMERPRO |
|---|---|---|---|---|---|
| Cartpole Swingup | $126 \pm 16$ | $332 \pm 66$ | $521 \pm 80$ | $479 \pm 45$ | $\mathbf{671 \pm 42}$ |
| Cheetah Run | $30 \pm 2$ | $334 \pm 17$ | $\mathbf{444 \pm 35}$ | $\mathbf{477 \pm 16}$ | $349 \pm 61$ |
| Cup Catch | $88 \pm 73$ | $\mathbf{553 \pm 60}$ | $477 \pm 175$ | $\mathbf{550 \pm 69}$ | $493 \pm 109$ |
| Finger Spin | $10 \pm 1$ | $629 \pm 207$ | $655 \pm 133$ | $511 \pm 115$ | $\mathbf{826 \pm 162}$ |
| Reacher Easy | $82 \pm 39$ | $400 \pm 296$ | $462 \pm 130$ | $\mathbf{614 \pm 164}$ | $\mathbf{641 \pm 123}$ |
| Walker Run | $35 \pm 4$ | $219 \pm 9$ | $161 \pm 6$ | $136 \pm 17$ | $\mathbf{394 \pm 33}$ |

Finger Spin and Reacher Easy, and demonstrates better data efficiency on Cup Catch. We notice a large variance in DREAMERPRO's performance on Finger Spin. Further investigation reveals that DREAMERPRO learned close to optimal behavior (with average episode returns above 950) on two of the seeds, while converged to a suboptimal behavior (with average episode returns around 500) on the other seed. The low variance of DREAMER indicates that it hardly achieved close to optimal behavior. Our results suggest for the first time that prototypical representations (and reconstruction-free representation learning in general) can be beneficial to MBRL even in the absence of strong visual distractions.

## 4.2 PERFORMANCE IN NATURAL BACKGROUND DMC

Figure 3 and Table 2 respectively show the performance curves and final evaluation returns obtained by all models in the natural background setting. DREAMER completely fails on all tasks, showing the inability of reconstruction-based representation learning to deal with complex visual distractions. In contrast, DREAMERPRO achieves the best performance on 4 out of 6 tasks, with large performance gains from baselines on Cartpole Swingup, Finger Spin, and Walker Run. We additionally train TPC with a batch size of 300 (denoted TPC-Batch-300), and DREAMERPRO is still able to outperform it on 4 out of 6 tasks. These results indicate that the advantage of prototypical representations over contrastive learning in computer vision can indeed be transferred to MBRL for better robustness to visual distractions.

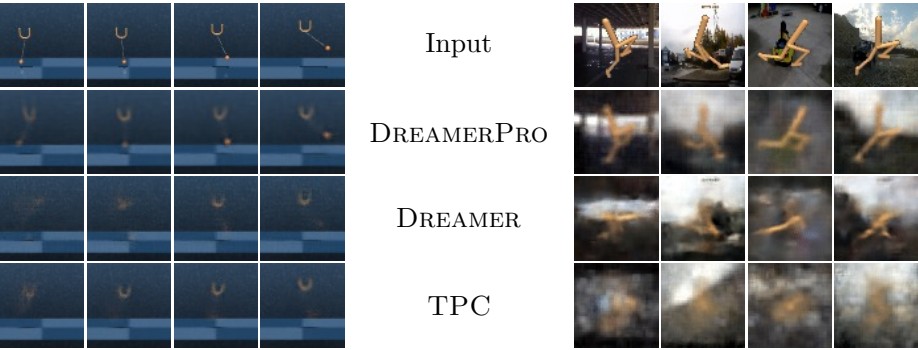

Figure 4: Visualization of learned latent states through reconstruction from an auxiliary decoder. The first row shows the input images, and the remaining rows show the reconstruction from the RSSM state $z_t$ for each model.

### 4.3 VISUALIZATION AND ANALYSIS

To better understand how the model works and explain the performance gaps, we visualize the learned latent states through reconstruction from an auxiliary decoder (Figure 4).

Figure 4 (Left) shows the reconstructions for Cup Catch after 100K environment steps. Note that the reconstruction of the ball is only possible from the states learned by DREAMERPRO, explaining its better data efficiency. DREAMER fails to reconstruct the ball at this early stage, probably because the ball takes only a few pixels, lowering its priority in the reconstruction loss.

Figure 4 (Right) shows the reconstructions for Walker Run after 1M environment steps. We see that both DREAMER and DREAMERPRO capture some information of the background, but only DREAMERPRO is able to recover the posture of the Walker. The reconstruction from TPC does not seem relevant to the Walker or the background. This indicates that while TPC may be better at discarding distractors, DREAMERPRO is better at retaining task-relevant information.

### 4.4 ABLATION STUDY

We now show the individual effect of the two loss terms, $\mathcal{J}_{\text{SwAV}}^t$ and $\mathcal{J}_{\text{Temp}}^t$, in Figure 6. Here, each of the ablated versions, DreamerPro-No-SwAV and DreamerPro-No-Temp, removes one of the loss terms. We did not investigate removing both terms, as its failure has been shown in DREAMER (Hafner et al., 2020). We train the ablated versions in natural background DMC, and observe that both terms are necessary for achieving good performance. In particular, naively combining SwAV with DREAMER (i.e., DreamerPro-No-Temp) leads to inferior performance, as it ignores the temporal structure. On the other hand, $\mathcal{J}_{\text{Temp}}^t$ alone is not sufficient to provide meaningful cluster assignment targets and learning signals for the convolutional encoder.

### 5 RELATED WORK

**Self-supervised representation learning for static images.** Recent works in self-supervised learning have shown its effectiveness in learning representations from high-dimensional data. CPC (van den Oord et al., 2019) learns representations by maximizing the mutual information between the encoded representations and its future prediction using noise-contrastive estimation. SimCLR (Chen et al., 2020) shows that the contrastive data can be generated using the data in the training mini-batch by applying random augmentations. MoCo (He et al., 2020), on the other hand, improves the contrastive training by generating the representations from a momentum encoder instead of the trained network. Despite the success in some tasks, one weakness of the contrastive approaches is that it require the model to compare a larger amount of samples, which demands large batch sizes or memory banks. To address this problem, some works propose to learn the image representations without discriminating between samples. Particularly, BYOL (Grill et al.,

2020) introduces a momentum encoder to provide target representations for the training network. SwAV (Caron et al., 2020) proposes to learn the embeddings by matching them to a set of learned clusters. DINO (Caron et al., 2021) replaces the clusters in SwAV with categorical heads and uses the centering and sharpening technique to prevent representations collapsing. Unlike our model, these works treat each image independently and ignore the temporal structure of the environment, which is crucial in learning the forward dynamics and policy in MBRL.

**Representation learning for model-free reinforcement learning.** It has been shown that adopting data augmentation techniques like random shifts in the observation space enables robust learning from pixel input in any model-free reinforcement learning algorithm (Laskin et al., 2020b; Yarats et al., 2021c;a). Recent works have also shown that self-supervised representation learning techniques can bring significant improvement to reinforcement learning methods. For example, CURL (Laskin et al., 2020a) performs contrastive learning along with off-policy RL algorithms and shows that it significantly improves sample-efficiency and model performance over pixel-based methods. Other works aim to improve the representation learning quality by combining temporal prediction models in the representation learning process (Schwarzer et al., 2021a;b; Stooke et al., 2021; Yarats et al., 2021b; Guo et al., 2020; Gregor et al., 2019). However, the main purpose of the temporal prediction models in these works is mainly to obtain the abstract representations of the observations, and they are not shown to support long-horizon imagination.

**Model-based reinforcement learning with reconstruction.** Model based reinforcement learning from raw pixel data can learn the representation space by minimizing the observation reconstruction loss. World Models (Ha & Schmidhuber, 2018) learn the latent dynamics of the environment in a two-stage process to evolve their linear controllers in imagination. SOLAR (Zhang et al., 2019) models the dynamics as time-varying linear-Gaussian and solves robotic tasks via guided policy search. Dreamer (Hafner et al., 2020) jointly learns the RSSM and latent state space from observation reconstruction loss. DeepMDP (Gelada et al., 2019) also propose a latent dynamics model-based method that uses bisimulation metrics and reconstruction loss in Atari. However, reconstruction based methods are susceptible to noise and objects irrelevant to the task in the environment (Nguyen et al., 2021). Furthermore, in a few cases, the latent representation fails to reconstruct small task-relevant objects in the environment (Okada & Taniguchi, 2021).

**Reinforcement learning under visual distractions.** A large body of works on robust representation learning focuses on contrastive objectives. For example, CVRL (Ma et al., 2020) proposes to learn representations from complex observations by maximizing the mutual information between an image and its corresponding embedding using contrastive objectives. However, the learning objective of CVRL encourages the representation model to learn as much information as possible, including task-irrelevant information. Dreaming (Okada & Taniguchi, 2021) and TPC (Nguyen et al., 2021) tackle this problem by incorporating a dynamic model and applying contrastive learning in the temporal dimension, which encourages the model to capture controllable and predictable information in the latent space. Bisimulation metrics method such as DBC (Zhang et al., 2021) and PSE (Agarwal et al., 2021) is another type of representation learning robust to visual distractions. Using the bisimulation metrics that quantify the behavioral similarity between states, these methods make the mode robust to task-irrelevant information. However, DBC cannot generalize to unseen backgrounds (Nguyen et al., 2021), and PSE is only shown to work in the model-free setting and requires a pre-trained policy to compute the similarity metrics, while our model learns both the world model and the policy from scratch.

## 6 CONCLUSION

In this work, we presented the first reconstruction-free MBRL agent based on the prototypical representation and its temporal dynamics. In experiments, we demonstrated the consistently improved accuracy and robustness of the proposed model in comparison to the Temporal Predictive Coding (TPC) agent and the Dreamer agent for both standard and natural background DMC tasks. Our results suggest that there are unexplored broad areas in reconstruction-free MBRL. Interesting future directions are to apply this model on Atari games and to investigate the possibility of learning hierarchical structures such as skills without reconstruction.

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

## A    HYPERPARAMETERS

For hyperparameters that are shared with DREAMER, we use the default values suggested in the config file in the official implementation of DREAMER, with the following two exceptions. We set `rssm.discrete = False` and `actor.dist = tanh_normal`, as we find these changes improve performance over the default setting. The additional hyperparameters introduced in DREAMERPRO are listed in Table 3. We find it helpful to freeze the prototypes for the first 10K gradient updates. In the natural background setting, we add a squared loss that encourages the $\ell_2$-norm of projections (before $\ell_2$-normalization) to be close to 1. This helps stabilize the model.

Table 3: Additional hyperparameters in DREAMERPRO.

| Hyperparameter | Value |
| --- | --- |
| Number of prototypes $K$ | 2500 |
| Prototype dimension | 32 |
| Softmax temperature $\tau$ | 0.1 |
| Sinkhorn iterations | 3 |
| Sinkhorn epsilon | 0.0125 |
| Momentum update fraction $\eta$ | 0.05 |

## B  NEAREST NEIGHBOR QUERIES IN LATENT SPACE

Figure 5: Visualization of learned latent states through nearest neighbor queries.

We sample a batch of trajectories from the training replay buffer, and obtain the latent state for each image. Then, given a query image, we show the three images in the batch whose latent states are the closest to the query image. We use the same batch and same query images for all models.

For TPC, the nearest neighbors tend to contain different backgrounds, but the agent's states can also be very different (See Cup Catch and Walker Run). On the other hand, the nearest neighbors for DREAMERPRO tend to have similar backgrounds and also similar agent states.

Our results suggest that DREAMERPRO and TPC work in quite different ways. DREAMERPRO tries to retain task-relevant information at the cost of also including some distractors, while TPC focuses more on discarding distractors. An interesting future direction is to simultaneously consider these two factors and achieve a better balance.

## C    ABLATION RESULTS

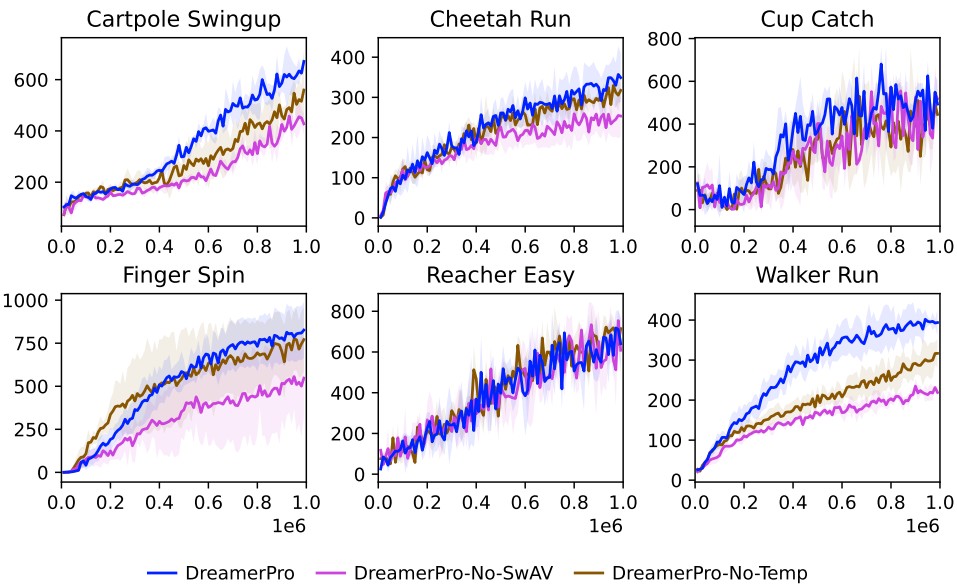

Figure 6: Ablation study. Both $\mathcal{J}^t_{\text{SwAV}}$ and $\mathcal{J}^t_{\text{Temp}}$ are necessary for achieving good performance.

## D    ATARI RESULTS

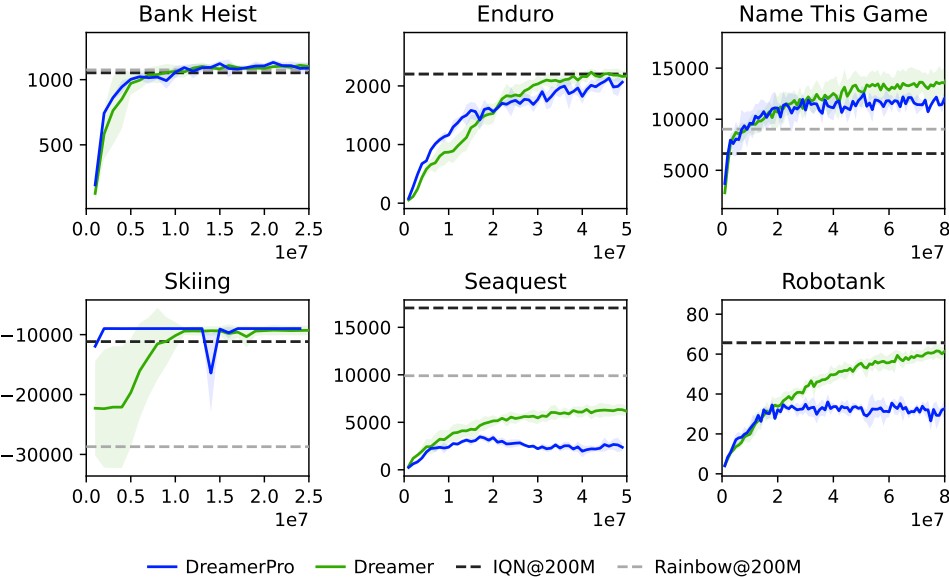

Figure 7: Performance curves in six Atari games.

To show the potential of DREAMERPRO to leverage the benefits of world models and discrete latents in complex environments, we train DREAMERPRO on a subset of six Atari games for 25M, 50M, and 80M environment steps depending on the convergence speed of DREAMER on these games. We freeze the prototypes for 30K gradient updates. The weights for reward and KL losses are 100 and 1, respectively. We additionally replace the linear projection layer from $z_t$ to $x_t$ by an MLP with one hidden layer of size 600. All other hyperparameters are kept as default.

On 4 out of the 6 games, DREAMERPRO obtains comparable performance to DREAMER, matching or surpassing model-free baselines that are trained for 200M environment steps. We note that this is the first time a reconstruction-free MBRL agent shows promising results on Atari with discrete latents.

# E  PROTOTYPE VISUALIZATIONS

We visualize the first 5 prototypes learned for each task using nearest neighbor queries, shown in each row of Figures 8 - 13. To do so, we sample a batch of trajectories from the training replay buffer, and obtain the projection $x_t$ from latent state $z_t$ for each image. Then, given a prototype $c_k$ as query, we show the ten images in the batch whose projections are the closest to the prototype.

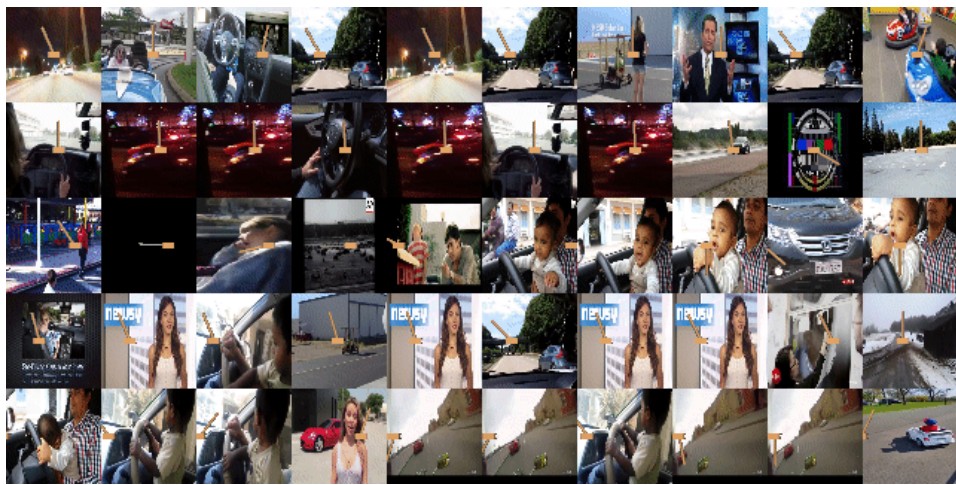

Figure 8: Prototype visualization for Cartpole Swingup.

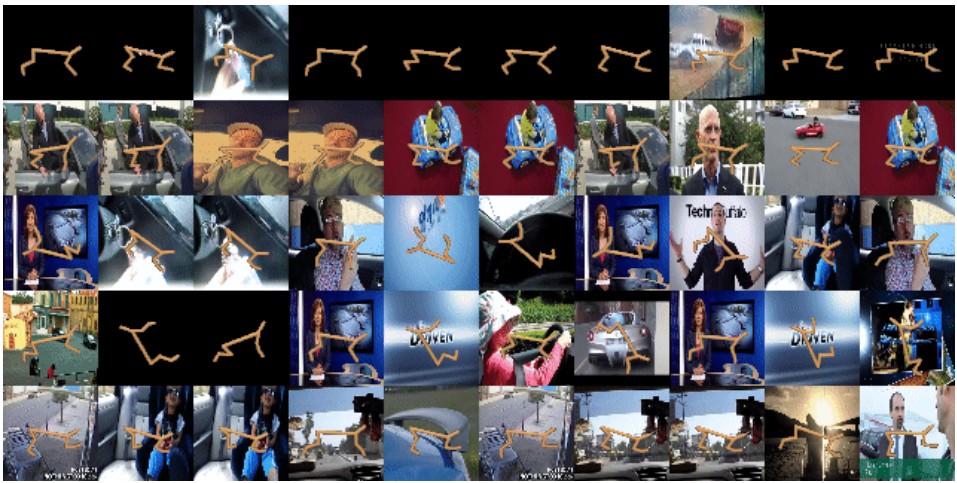

Figure 9: Prototype visualization for Cheetah Run.

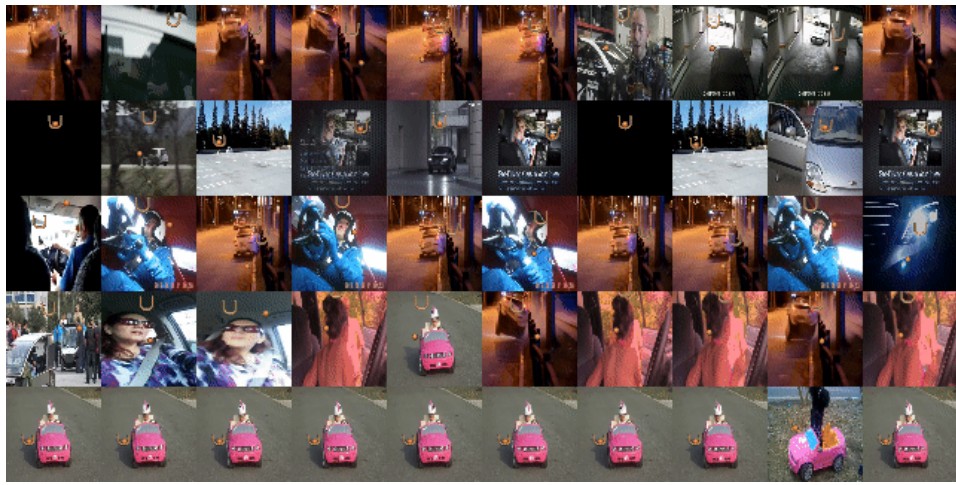

Figure 10: Prototype visualization for Cup Catch.

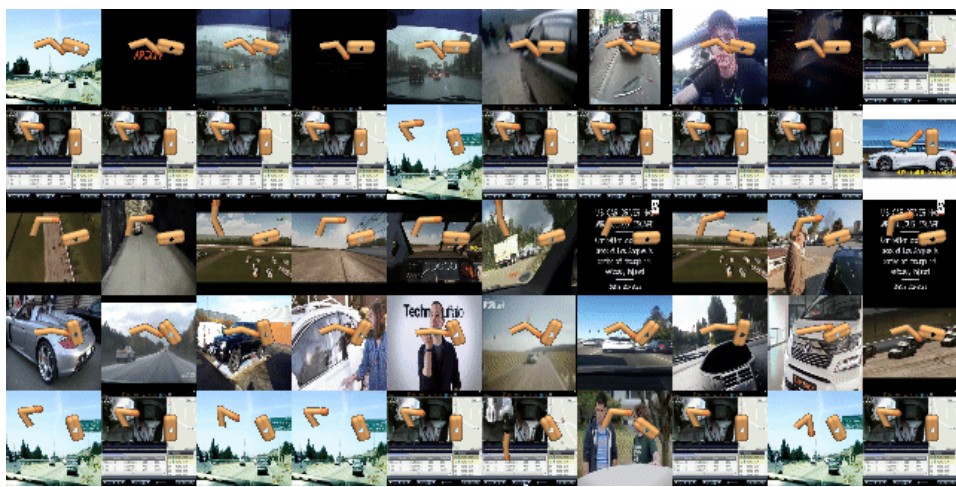

Figure 11: Prototype visualization for Finger Spin.

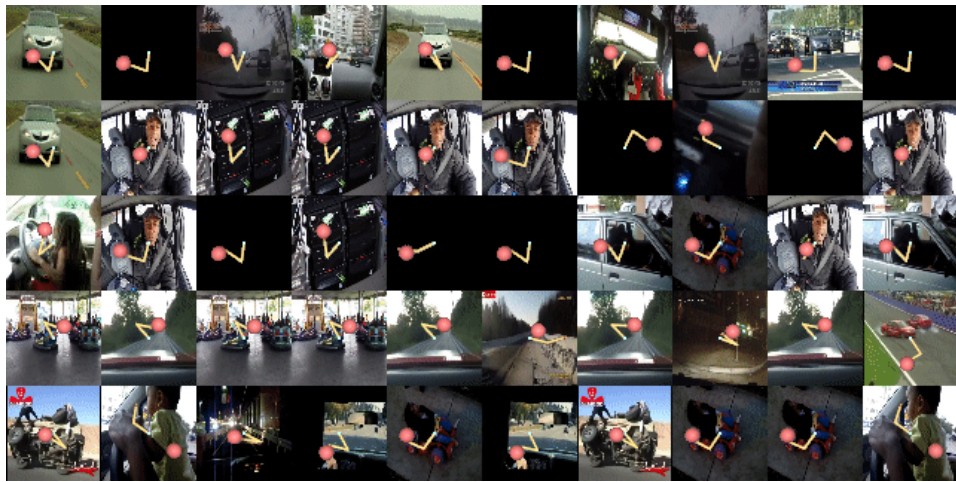

Figure 12: Prototype visualization for Reacher Easy.

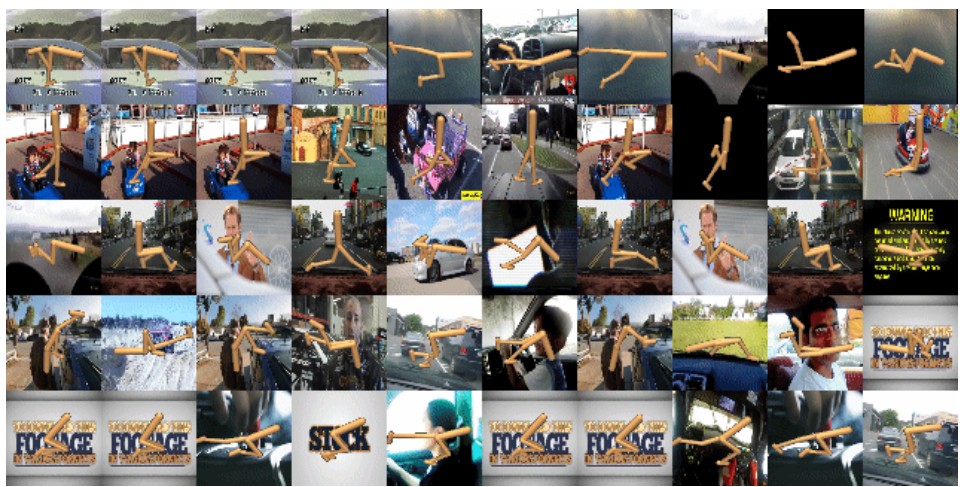

Figure 13: Prototype visualization for Walker Run.

