# OpenReview forum: "DreamerPro: Reconstruction-Free Model-Based Reinforcement Learning with Prototypical Representations"
_ICLR.cc/2022/Conference — ICLR 2022 Submitted_

### Official Review · Reviewer_VAKJ · 2021-10-21

**Correctness:** 4
**Technical Novelty And Significance:** 3
**Empirical Novelty And Significance:** 3
**Recommendation:** 5
**Confidence:** 4

**Main Review:**

Strengths
1. The paper is well-written and easy to follow.
2. The idea is straightforward and the experimental results are strong in DMC suite.
3. Introducing $\mathcal{L}_\text{temp}$ is a clean solution to integrating the temporal notion to the original SWAV objective.

Weaknesses
1. The major concern is the complexity of the environments. The results are evaluated in DMC, and in the 6 environments the proposed method outperforms DREAMER in terms of converging performance only in ReacherEasy. Further evaluation on Atari will make the results much more convincing.
2. The paper is not particularly strong in terms of novelty as swapping the reconstruction objective with prototypical objective is already shown to be effective in model-free RL (Yarats et al. 2021). This won't be a big problem if the paper can showcase effectiveness in more complex environments.


Minor
1. The batch size $B$ in all experiments seems not specified in the paper.


**Summary Of The Paper:**

This work proposes a reconstruction free method for pixel-based MBRL, utilizing cluster assignment adapted from SWAV with an additional temporal term. The method learns a latent space that discards task-irrelevant visual details and is more efficient compared to contrastive methods.

**Summary Of The Review:**

I think it is a very reasonable approach to combine cluster assignment to MBRL and the paper does a good job addressing the temporal consistency, a challenge that is not presented in model-free RL. However, I am not confident how the method could scale to higher complexity environments. I would be happy to increase my score if the above concern is addressed.

---

> ### Author Response · Authors · 2021-11-23
> **Response to Reviewer VAKJ**
>
> Thank you for your helpful comments. Please refer to the general response, where we showed Atari results, added new experiments that contribute to the understanding of DreamerPro and TPC, and clarified the batch size.

---

> > ### Comment · Reviewer_VAKJ · 2021-11-30
> > **Response to rebuttals**
> >
> > I thank the authors for the additional results on Atari and the clarification. While the proposed method is reasonably motivated, it requires a strong experimental results to be justified. DreamerPro is able to be comparable with Dreamer only on a subset of the Atari environments tested, and it remains unclear to me whether there is a fundamental limitation in the representation capacity of DreamerPro to handle complex environments. I would like to keep my original score.

---

### Official Review · Reviewer_xtkN · 2021-11-02

**Correctness:** 4
**Technical Novelty And Significance:** 3
**Empirical Novelty And Significance:** 2
**Recommendation:** 6
**Confidence:** 4

**Main Review:**

###  Strengths

- The approach is clear and well motivated, and a natural extension of representation learning for MBRL
- Related work section is thorough and well written
- Experimental results show an increased robustness to visual distractions compared to dreamer

###  Areas to improve

The experiments are not as thorough as they could be.
- Several baselines are cited but not included in experiments (ie PSE, CURL, many of the model-free temporal prediction models).  I understand that the paper is not attempting to contribute a model-free vs model-based comparison, but it makes the results seem thin.
- Three random seeds seems too few
- One of the main stated motivations of using a non-contrastive representation approach was to reduce the required batch size, but there is little evidence here with which to evaluate that claim.
- I like the ablation study comparing the different terms of the additional loss (Figure 4).  The paper would be stronger with some additional exploration of the design space of the presented idea.  For example, is the approach sensitive to the hyperparameters in Table 3?  Another example, for the temporal consistency loss, could you also include terms for enforcing the consistency across timesteps and augmentations simultaneously, eg w^(1)log u^(2)?

I found the diagram in Figure 1 a bit unclear because it ignores the augmentation index i, and has AUG(o_t) repeated twice.  What do the two augmentations in the diagram represent, if both sets of augmentations are actually passed through all paths of the diagram?


## Update after rebuttal

Thank you for addressing my concerns.  I will maintain my score that this paper is above the acceptance threshold.

**Summary Of The Paper:**

The paper presents DreamerPro, a method for learning a latent dynamics model from image observations without a pixel-based reconstruction loss.  The model combines the clustering approach of SwAV with an additional temporal consistency loss.  Experimental results show improved performance over baselines in distraction-free and natural-video-background settings on DeepMind Control Suite tasks.


**Summary Of The Review:**

Overall, I found the paper to be a novel exploration of a natural combination of two methods from MBRL and SSL.  The paper makes modest claims, and supports them with good evidence and a well-written explanation.  Although the results show empirical improvements, the paper could do more to contribute to a more general understanding of these improvements and the approach itself.

---

> ### Author Response · Authors · 2021-11-23
> **Response to Reviewer xtkN**
>
> Thank you for your positive and thoughtful comments. Please refer to the general response regarding baselines, batch size, and new results.
>
> > Three random seeds seems too few.
>
> We agree. Unfortunately we cannot afford to run more seeds at this time. Kindly note that TPC also uses three seeds.
>
> > Is the approach sensitive to the hyperparameters in Table 3?
>
> We find that the number of prototypes $K$ can make a difference. Smaller $K$ leads to worse performance. We tried prototype dimensions in {32, 64, 128}, and higher dimension did not improve performance. We did not tune the temperature $\tau$ and Sinkhorn iterations. We tried Sinkhorn epsilon in {0.00625, 0.0125, 0.025, 0.05}, and they tend to have similar performance. The momentum values in {0.005, 0.01, 0.05} also seem to give similar performance.
>
> > Could you also include terms for enforcing the consistency across timesteps and augmentations simultaneously?
>
> Thanks for your suggestion. We tried this before, and it did not improve performance. We monitored the consistency loss for same and different augmentations, and we found that optimizing the consistency for the same augmentation will also improve the consistency for different augmentations.
>
> > Figure 1 a bit unclear, What do the two augmentations in the diagram represent?
>
> Thanks for pointing this out. We have clarified that each AUG(o_t) is an independent application of augmentation.

---

### Official Review · Reviewer_FvUb · 2021-11-02

**Correctness:** 3
**Technical Novelty And Significance:** 2
**Empirical Novelty And Significance:** 3
**Recommendation:** 5
**Confidence:** 3

**Main Review:**

=== Strengths ===

+ The paper is well written and motivated, and the presentation is easy to follow.
+ The presented approach shows convincing experimental results on the selected benchmark.

=== Weaknesses ===

- I would like to see a more thorough experimental analysis. The latest Dreamer implementation also evaluates all tasks in the Atari benchmark. I would like to see how DreamerPro compares to the basic Dreamer on the Atari benchmark as well.
- I would like to see some visualizations on the learned features using the prototype learning loss. For example, a side-by-side comparison of t-SNE or nearest neighbor plots of the image observations with their representations learned from 1) the original reconstruction loss and 2) the proposed prototype learning loss. I would also like to visualize what the learned prototypes correspond to in the observation space.
- The authors mention that they use continuous latent space for both Dreamer and DreamerPro as they found it to perform better. However, according to Dreamerv2, discrete latent space is one of the major improvements they found and it outperformed continuous latent space. I would appreciate it if the authors can further clarify this in the response.
- I am concerned about the novelty of this paper, because it does seem like a straightforward combination of Dreamer and SWAV, with one additional novelty which is the proposed L_temp loss. This feels somewhat incremental to me.

### Post rebuttal updates
The authors did a good job in rebuttal and with the additional experiments and visualizations, the paper is now in a much better shape in my honest opinion. I am updating my score from 3 to 5 to reflect the changes. While the paper proposes a valid combination of existing techniques (SwAV + Dreamer) with reasonable performance on the selected task of "RL with random visual distractions", to me this paper is not significant or novel enough to make a strong case for acceptance.

**Summary Of The Paper:**

This paper proposes DreamerPro, an extension to Dreamer where it swaps the reconstruction loss with a prototype learning loss based on SWAV. At each optimization step, DreamerPro augments each batch of observations into two using different augmentations. It uses the Sinkhorn-Knopp algorithm to compute the target clusters for each batch, with the goal of   encouraging uniform clustering assignment within each augmented batch. Intuitively, the prototype learning loss rewards prediction accuracy both from across the views, which adapts from the original SWAV approach, and within the same view, which the authors mention help build temporal information into the learned prototypes. DreamerPro outperforms the original Dreamer on most of the tasks on the Deepmind Control benchmark, and drastically improves on the Deepmind Control benchmark with noisy backgrounds.


**Summary Of The Review:**

The paper is well-written and the presented approach shows promising results on the selected benchmark. However, it seems that the paper lacks novelty at the moment, and I would like to see a more thorough experimental section with more visualizations. Based on these, I do not recommend acceptance at this moment.

---

> ### Author Response · Authors · 2021-11-23
> **Response to Reviewer FvUb**
>
> Thank you for your suggestions for improvement. Please refer to the general response for Atari experiments,  visualizations and novelty.
>
> > Continuous vs discrete latents.
>
> DreamerV2 showed the benefit of discrete latents in Atari. However, in DMC, we find continuous latents to work slightly better, possibly because reparameterization gradients are used for DMC, and continuous latents can better backpropagate the gradients. Our Dreamer results are comparable to those reported in [DrQ-v2](https://arxiv.org/pdf/2107.09645.pdf).

---

> > ### Comment · Reviewer_FvUb · 2021-11-29
> > **Response**
> >
> > Thank you for the response, I have updated my score and review to reflect the latest changes.

---

### Official Review · Reviewer_pipU · 2021-11-02

**Correctness:** 2
**Technical Novelty And Significance:** 2
**Empirical Novelty And Significance:** 2
**Recommendation:** 3
**Confidence:** 4

**Main Review:**

The core novelty of the paper is to combine Dreamer approach to model-based reinforcement learning in pixel space with prototypical based method for learning representations from SwAV. This led to a method that can learn a good enough representation for planning, without reconstructing the observed images. Removing reconstruction is generally well motivated, however, the paper fails to support some of its key claims.

1. Unclear motivation and focus. At the very high level, the paper mentions two limitations for reconstruction-based methods (in intro). First, they are computationally expensive, and second, they learn signals irrelevant to the task. In the results, however, the paper entirely ignores one and focuses on the second i.e. there is no talk or analysis on the computational aspect of the proposed method and it seems the authors' main focus was to get better results on environment with random video backgrounds.

2. Weak baselines, particularly for contrastive methods. The paper is quite vocal about the limitations of contrastive representation learning and why the authors chose to use prototypical based method instead. However, the only contrastive baseline is TPC which is not representative of this group of models. e.g. the Dreamer paper itself also proposes a contrastive loss for training the latent model although it is not temporal. There is also Dreaming (https://arxiv.org/abs/2007.14535) and CVRL (https://arxiv.org/abs/2008.02430). Moreover, contrastive and prototypical representation learning are not the only solutions to this problem e.g. there are already existing bisimulation methods which address this exact problem (https://arxiv.org/abs/2006.10742).

3. Lack of analysis. Unfortunately, the paper does not provide any analysis on *how* the model outperforms Dreamer. More strangely, the paper claims better sample efficiency vs Dreamer even in absence of distraction. This is strange because reconstruction of a basic static background does not "waste the representation capacity" as argued by the author. Intuitively, it can be assumed that it is mainly because the model learnt a more robust representation, however, there is no explicit theory or experiment to support this claim. For examples of such analysis please refer to Fig 5a in TPC paper.

===== Other issues
1. The ablation study (Fig 4) is poorly described. It's not even clear which set of environments (distracting or not distracting) the model is tested on and why a particular subset of original six tasks is visualized.
2. The first reference to SwAV (3rd paragraph of page 2) is not cited.
3. In page 4, it is mentioned that the number of prototypes K is set to B x T. but it is not clear what T is. Is T the time step? So the number of prototypes grow by the time and the model is approximating millions of prototypes at the end?
4. The reported numbers for TPC does not match the reported numbers in the original paper. For example, on Walker Run, TPC reported ~400 while this paper reported ~200. Is this because of the re-implementation?

**Summary Of The Paper:**

The paper proposes DreamerPro, a clustering based (i.e. prototype-based) version of Dreamer, which learns the latent representations by prototypical representation learning instead of image reconstruction. In other words, the proposed method is Dreamer without reconstruction replaced by SwAV template prototypical method. The authors test their method on a subset of six DeepMind Control tasks from pixels, with an without distracting backgrounds, to demonstrate that it is capable of outperforming Dreamer and TPC in almost all cases.

**Summary Of The Review:**

The proposed method is a combination of two previously established methods, Dreamer and SwAV. This combination and its applications in MBRL is novel, however, I don't find it to be particularly significant. Reconstruction free MBRL from pixels is not a new problem and the paper does not provide enough theoretical (nor empirical) insights as why a prototype-based method is a better fit for such setting vs other methods.

---

> ### Author Response · Authors · 2021-11-23
> **Response to Reviewer pipU**
>
> We appreciate your constructive comments. For addtional baselines and analysis, please refer to the general response.
>
> > No talk or analysis on the computational aspect of the proposed method.
>
> We agree that reducing the computation cost induced by reconstruction is a motivation, but not the main focus of this paper. We believe that the computational benefit of reconstruction-free methods is most signficant when the observations are high-resolution images. However, creating and training in such environments would require significantly more resource. So before going there, it is reasonable to first focus on task performance in low-resolution environments.
>
> > Unclear description of ablation study.
>
> Thanks for pointing this out. We have clarified that the ablation study is done in natural background DMC, and we now include all six tasks.
>
> > The first reference to SwAV is not cited.
>
> Fixed. Thanks.
>
> > What is $T$?
>
> $T$ is the length of the observation sequence $o_{1:T}$ sampled from the replay buffer. $T$ is a constant and the default value is 50.
>
> > Walker Run performance in standard DMC is lower than original TPC.
>
> Yes, this can happen with re-implementation. We note that (1) this does not change our conclusion because DreamerPro is a lot better on this task, and (2) our re-implementation is better than the original version on many tasks in natural background DMC, e.g., Cartpole Swingup: ~500 (ours) vs ~400 (original), Cup Catch: ~500 (ours) vs ~0 (original), Walker Run: ~150 (ours) vs ~100 (original).

---

> > ### Comment · Reviewer_pipU · 2021-11-30
> > **Response to the response.**
> >
> > Thank you for your response. I believe your comment addressed one of the major issues (motivation). I agree that resource can be an issue in this case. Unfortunately. it does not address the reminder of the issues (weak baselines and lack of analysis). This paper has the potential to be much more impactful, if compared if stronger baselines and particularly if it provides more analysis and insights. That's why, in its current form, I have to stick to my current recommendation.

---

> > > ### Author Response · Authors · 2021-11-30
> > > **Could you please check the general response?**
> > >
> > > Thanks for your comment. We tried to address your concerns about baselines and analysis in the general response titled **To All Reviewers**. We would appreciate it if you could also check that out and let us know your feedback.
> > >
> > > Below is a brief summary:
> > > 1. We added Dreaming as a baseline. DreamerPro outperforms Dreaming on 5 out of 6 tasks in natural background DMC (Table 2). We did not compare to DBC and CVRL because we use the same background distractions as TPC, and TPC already outperforms DBC and CVRL on most tasks (and thus should be considered a strong baseline).
> > > 2. Our visualization showed that DreamerPro is better at retaining task-relevant information (Figure 4). For Cup Catch in standard setting, the ball can be reconstructed from the latent states learned by DreamerPro after only 100K steps, which is not the case for Dreamer and TPC. For Walker Run in natural background setting, only DreamerPro can recover the approximate pose of the Walker.

---

### Author Response · Authors · 2021-11-23
**To All Reviewers (2/2)**

**More baselines (pipU, xtkN)**

We added Dreaming as an additional baseline for DMC experiments (updated Figures 2,3 and Tables 1,2). We also compared to IQN and Rainbow, two model-free baselines for Atari experiments. Kindly note that TPC matches or outperforms CVRL, DBC, and CURL on most tasks (See Figures 2,3,7 in TPC paper). We use the same background distractions as TPC, and DreamerPro in turn matches or outperforms TPC on most tasks. Hence, we did not include CVRL, DBC, and CURL as baselines.

**Batch size and computation cost (pipU, xtkN, VAKJ)**

For Dreamer, Dreaming, and DreamerPro, we used the default batch size, which is 50. Note that Dreaming and DreamerPro both use data augmentation, so the effective batch size is 100. For TPC, we used batch size 150, so that the wall clock training time is similar to DreamerPro. We used mixed precision for DreamerPro and TPC, but found it necessary to use float32 for Dreaming, resulting in its longer wall clock training time.

Our main result in natural background DMC (Table 2) suggests that with similar computation budget, DreamerPro matches or outperforms Dreaming and TPC each on 5 out of 6 tasks. We additionally trained TPC with batch size 300, and DreamerPro (with batch size 50) is still able to outperform it on 4 out of 6 tasks.

**Novelty (FvUb, VAKJ)**

We agree that our model is not entirely new, but a novel combination of existing work. We believe that our new experiments have added to the novelty. In particular, our visualizations revealed the different working modes of DreamerPro and TPC, suggesting that future work should consider retaining task-relevant information and discarding distractors simultaneously. Our Atari experiments demonstrate for the first time that reconstruction-free MBRL agents have the potential to learn world models with discrete latents in complex environments.

---

### Author Response · Authors · 2021-11-23
**To All Reviewers (1/2)**

We thank all reviewers for their time and insightful feedback. In this general response, we would like to address the common concerns from reviewers.

**Visualization and analysis (pipU, FvUb)**

We added visualization of learned latent states, including reconstruction from an auxiliary decoder (Figure 4) and nearest neighbor queries (Appendix Figure 5). We also visualize the learned prototypes through nearest neighbor queries (Appendix Figures 8-13).

For both DreamerPro and TPC, the auxiliary decoder is trained along with the agent learning process to reconstruct the input image from RSSM state $z_t$, and $z_t$ does not receive gradients from the decoder.

Figure 4 (Left) shows the reconstructions for Cup Catch after 100K environment steps. Note that the reconstruction of the ball is only possible from the states learned by DreamerPro, explaining its better data efficiency. Dreamer fails to reconstruct the ball at this early stage, probably because the ball takes only a few pixels, lowering its priority in the reconstruction loss.

Figure 4 (Right) shows the reconstructions for Walker Run after 1M environment steps. We see that both Dreamer and DreamerPro capture some information of the background, but only DreamerPro is able to recover the posture of the Walker. The reconstruction from TPC does not seem relevant to the Walker or the background. This indicates that while TPC may be better at discarding distractors, DreamerPro is better at retaining task-relevant information.

Similar results can be seen in the nearest neighbor queries. For TPC, the nearest neighbors tend to contain different backgrounds, but the agent's states can also be very different (See Finger Spin and Walker Run in Appendix Figure 5). On the other hand, the nearest neighbors for DreamerPro tend to have similar backgrounds and also similar agent states.

Our results suggest that DreamerPro and TPC work in quite different ways. DreamerPro tries to retain task-relevant information at the cost of also including some distractors, while TPC focuses more on discarding distractors. An interesting future direction is to achieve a better balance.

**Atari experiments (FvUb, VAKJ)**

We added Atari results in Appendix Figure 7.

Our main focus is to improve robustness to distractions. The observations from Atari games are not designed to distract the player. Therefore, we cannot expect to outperform Dreamer on Atari. Also, previous reconstruction-free methods did not consider Atari to be a benchmark for evaluating robustness to distractions.

Hence, our goal for the Atari experiments is to show the potential of DreamerPro to leverage the benefits of world models and discrete latents in complex environments. We note that the dynamics smoothing trick in TPC is only applicable to continuous latents, while DreamerPro can use both continuous and discrete latents. We trained DreamerPro using discrete latents on 6 Atari games for 25M, 50M, and 80M environment steps depending on the convergence speed of Dreamer on these games. The results are averaged over three independent runs. Unfortunately, we cannot afford longer training times or more seeds during the rebuttal. On 4 out of the 6 tasks, DreamerPro obtained comparable performance to Dreamer, matching or surpassing model-free baselines that were trained for 200M environment steps. We note that this is the first time a reconstruction-free MBRL agent shows promising results on Atari.

---

### Decision · Program_Chairs · 2022-01-20

**Decision:**

Reject

**Comment:**

The authors propose an alteration to Dreamer that incorporates a swav-like objective. The reviewers raised a number of issues with the paper, overall arguing for rejection. In particular, the reviewers felt that the work was not well motivated, weak performance, that a number of baselines were missing, and a lack of analysis of the results, a lack of novelty. While the authors addressed many of these concerns during the rebuttal, the majority of reviewers still felt this was not enough and that the paper did not meet the bar for acceptance. Therefore, I recommend rejection at this stage so that these concerns can be addressed.